# Influence of Iron Plaque on Accumulation and Translocation of Cadmium by Rice Seedlings

Abu Bakkar Siddique [1,2,3], Mohammad Mahmudur Rahman [1,2,*], Mohammad Rafiqul Islam [1,4], Muhammad Tahir Shehzad [5], Bibhash Nath [6] and Ravi Naidu [1,2]

1    Global Centre for Environmental Remediation (GCER), College of Engineering, Science and Environment, The University of Newcastle, Callaghan, NSW 2308, Australia; AbuBakkar.Siddique@uon.edu.au (A.B.S.); rafiqss69@bau.edu.bd (M.R.I.); ravi.naidu@newcastle.edu.au (R.N.)

2    Cooperative Research Centre for Contamination Assessment and Remediation of the Environment (CRC CARE), The University of Newcastle, Callaghan, NSW 2308, Australia

3    Department of Agriculture, Noakhali Science and Technology University (NSTU), Noakhali 3814, Bangladesh

4    Department of Soil Science, Bangladesh Agricultural University (BAU), Mymensingh 2202, Bangladesh

5    Institute of Soil and Environmental Science, University of Agriculture Faisalabad, Faisalabad 38040, Pakistan; Muhammad.shehzad@uaf.edu.pk

6    Department of Geography and Environmental Science, Hunter College of the City University of New York, New York, NY 10021, USA; bibhash.nath86@myhunter.cuny.edu

*    Correspondence: mahmud.rahman@newcastle.edu.au

**Abstract:** This study investigated the impact of soil type and rice cultivars on variations in the iron plaque formation and cadmium (Cd) accumulation by different portions of rice seedlings under the influence of Fe amendment. The experiments were performed in pots under glasshouse conditions using two typical paddy soils. Rice seedlings were exposed to three concentrations of Cd (0, 1 and 3 mg kg$^{-1}$ soil) and Fe (0, 1.0 and 2.0 g kg$^{-1}$ soil). The results revealed that shoot biomass decreased by 12.2–23.2% in Quest and 12.8–30.8% in Langi in the Cd1.0 and Cd3.0 treatments, while shoot biomass increased by 11.2–19.5% in Quest and 26–43.3% in Langi in Fe1.0 and Fe2.0 as compared to the Fe control. The Cd concentration in the roots and shoots of rice seedlings were in the order of Langi cultivar > Quest cultivar, but the Fe concentration in rice tissues showed the reverse order. Fe plaque formations were promoted by Fe application, which was 7.8 and 10.4 times higher at 1 and 2 g kg$^{-1}$ Fe applications compared to the control Fe treatment. The Quest cultivar exhibited 13% higher iron plaque formation capacity compared to the Langi cultivar in both soil types. These results indicate that enhanced iron plaque formation on the root surface was crucial to reduce the Cd concentration in rice plants, which could be an effective strategy to regulate grain Cd accumulation in rice plants.

**Keywords:** cadmium; iron; rice cultivar; soil type; CBD extracts

## 1. Introduction

Cadmium (Cd) is one of the most perilous and ubiquitous heavy metals in agricultural soils. Cd is categorized as a class I carcinogen and ranked 7th in the list of the top 10 toxic elements [1,2]. Being a non-essential trace element, Cd does not perform any known biological functions in plants and has received increased attention for safe crop production worldwide [3,4]. Industrial effluents and emissions, sewage sludge, phosphate fertilization, wastewater irrigation, pesticides use, and coal combustion residues are the potential anthropogenic sources that continually introduce Cd into paddy soils [5–8]. In addition, parent materials, mines, volcanos etc. are the natural sources of Cd input in paddy soils [3]. Relative to other heavy metals, Cd is readily mobile and soluble in soil [9] and is thus quickly taken up through the root systems [10]. After root uptake, Cd is conglomerated in different edible parts of the plant and does harm to living cells at very low concentrations [5,11,12]. The Cd levels in Australian soils are very low compared to China

and other Asian countries. For example, concentrations of Cd in cultivated soils ranged from 0.11 to 6.37 mg kg$^{-1}$ (median: 0.85 mg kg$^{-1}$ and mean: 1.3 mg kg$^{-1}$), whereas, in the background (unfertilised) soil, Cd ranged from 0.02 to 1.99 mg kg$^{-1}$ (median: 0.04 mg kg$^{-1}$ and mean: 0.36 mg kg$^{-1}$) in the Sydney region [13]. The ANZ environmental investigation guideline for soil Cd is 3 mg kg$^{-1}$ [14].

Rice is the most important cereal crop and a dominant staple food for more than half of the world's population [15]. It was found that the rice plant accumulates Cd from the soil in the form of $Cd^{2+}$, and after passing several processes, Cd is finally deposited into grains [10]. The Cd toxicity in growing rice plants displayed visible symptoms of growth inhibition in terms of biomass reduction in the roots and shoots, chlorosis, necrosis, and eventually the death of plants [16–18]. The notorious Itai-Itai disease that occurred in Japan during the mid-twentieth century, which emanated from the excessive intake of Cd-vitiated rice has brought public attention worldwide [19]. High Cd accumulation in rice has delivered Cd into the human body via food chain disruption, thus introducing several adverse health implications such as anaemia, cancer, cardiac failure, cerebrovascular infarction, emphysema, hypertension, lung damage, osteoporosis, proteinuria, and renal dysfunction, etc. [20]. The maximum permissible limit of Cd in rice established by the Chinese National Health Commission is 0.2 mg kg$^{-1}$ [21].

Rice plants exhibit similar characteristic features to wetland plants, including the formation of extensive aerenchyma that transfers $O_2$ from the shoots to roots during respiration activity [22]. To adapt to adverse anaerobic environment, the roots aerenchyma of rice plant released $O_2$ in the rhizosphere resulting in the deposition of Fe plaque on the root surfaces through the oxidation of $Fe^{2+}$ (ferrous iron) to $Fe^{3+}$ (ferric iron) [23]. Iron plaques mainly exist in the form of ferrihydrite ($5Fe_2O_3.9H_2O$), goethite ($\alpha$-FeOOH), and lepidocrocite ($\gamma$-FeOOH) on the root surfaces [24]. Iron plaque deposits possess ample functional groups with many sorption sites that are capable of binding metal ions in the root zone [25]. Because of the strong affinity of Fe oxyhydroxides to sequester metals through adsorption or co-precipitation, iron plaque on the root surfaces thus interferes with the metal availability in the root zone and impacts their accumulation and translocation in plants [25,26]. Soil mineral concentration, soil amendments, and rice cultivar influence the production of Fe plaque, and its potential to sequester contaminating metals [8]. The Fe amendments significantly increased the deposition of iron plaque and adsorption of Cd in iron plaque but Cd adsorption in iron plaque is reduced with the increasing Cd concentration in soil [27]. The application of $FeSO_4$ enhanced the root plaque deposition, which safeguarded the rice plant from Cd toxicity and Fe deficiency through increasing Fe concentration in rice plants [28]. This is confirmed by the plaque-induced plant in terms of boosted biomass production, chlorophyll content and photosynthetic efficacy. A recent investigation had revealed that iron plaque deposition capabilities and Cd sequestration in iron plaque, as well as Cd delivery in the upper parts of rice plants were varied among the five cultivars grown under hydroponic conditions [18]. Liu et al. (2016) showed that rice cultivars having high radial oxygen loss (ROL) had a higher ability to form root plaque, thereby decreasing Cd uptake and translocation from the root to the edible parts when amended with Fe-rich amendments [29]. However, most studies have investigated the effect of exogenous Fe application on iron plaque formation, Cd sequestration in iron plaque, and the remobilization and transportation of Cd in the aboveground parts under hydroponic conditions. Soil experiments provide a more comprehensive evaluation of iron plaque formation on Cd translocation using different types of rice cultivars cultivated in diverse Cd-polluted soil, as compared to hydroponic studies.

In this study, a pot experiment was conducted under glasshouse conditions utilizing two different sets of soils spiked with two levels of Cd and Fe along with two Australian rice cultivars. Therefore, the core aims of this study were to investigate (i) the Cd and Fe concentration in rice plants and iron plaque when exposed to different levels of Cd and Fe, (ii) to investigate the protective role of iron plaque formation in reducing Cd

accumulation through Fe application, and (iii) the influence of rice cultivar and soil types on Cd accumulation.

## 2. Materials and Methods

### 2.1. Soil Collection, Processing and Preparation

Two typical topsoils (0–15 cm, low pH red soil and neutral pH grey soil) were sampled from the uncontaminated lands of Yanco Agricultural Institute and Leeton Field Station, New South Wales (NSW), Australia, respectively, and transported to the soil laboratory for further processing. Each soil (about 1 ton) was collected from a single sampling point at 4 m × 4 m grid area. The soil samples were crushed and made free from root debris and other impurities. After being air-dried at room temperature, the samples were sieved using 4-mm stainless steel mesh-sieve, mixed thoroughly, and homogenised for the pot experiment. Air-dried subsamples (500 g) were further ground to less than 2 mm in diameter to determine the initial physicochemical properties of soils. The soil textural class was clay, and pH (soil:water = 1:10), electrical conductivity (EC), cation exchange capacity (CEC), organic carbon, total nitrogen, total As, total Fe, and total Cd were 4.6, 1638 (dS/cm), 9.5 (meq 100/g), 2.2%, 0.2%, 0.56 (mg kg$^{-1}$), 618 (mg kg$^{-1}$), and 0.32 (mg kg$^{-1}$) for red soil, and 6.6, 1580 (dS/cm), 11.2 (meq 100/g), 1.3%, 0.1%, 0.21 (mg kg$^{-1}$), 140 (mg kg$^{-1}$), and 0.13 (mg kg$^{-1}$) for grey soil, respectively.

### 2.2. Pot Experiment Design and Rice Cultivars Used

The entire experimental procedure is presented in Scheme 1. The pot culture experiment was conducted under glasshouse conditions of the Global Centre for Environmental Remediation (GCER) situated at Newcastle Institute for Energy and Resources (NIER) S-block at the University of Newcastle, Australia, from April to August 2018 under controlled environmental conditions. The day and night temperatures were maintained at 28 °C and 22 °C, respectively. The relative humidity was 70–80%, and light exposure varied between 12–14 h/day throughout the entire growth period. The pot used in this study was 9-L rectangular plastic pot (dimensions: length 31 cm, width 22 cm, and height 18 cm). Each plastic pot received 6 kg of processed soil (red or grey soil). The pots were then arranged in a Randomized Complete Block Design (RCBD) with three replicates. Two commonly grown Australian rice cultivars (Langi and Quest) were used as test crops in this study, which were collected from the DPI (Department of Primary Industries), NSW, Australia.

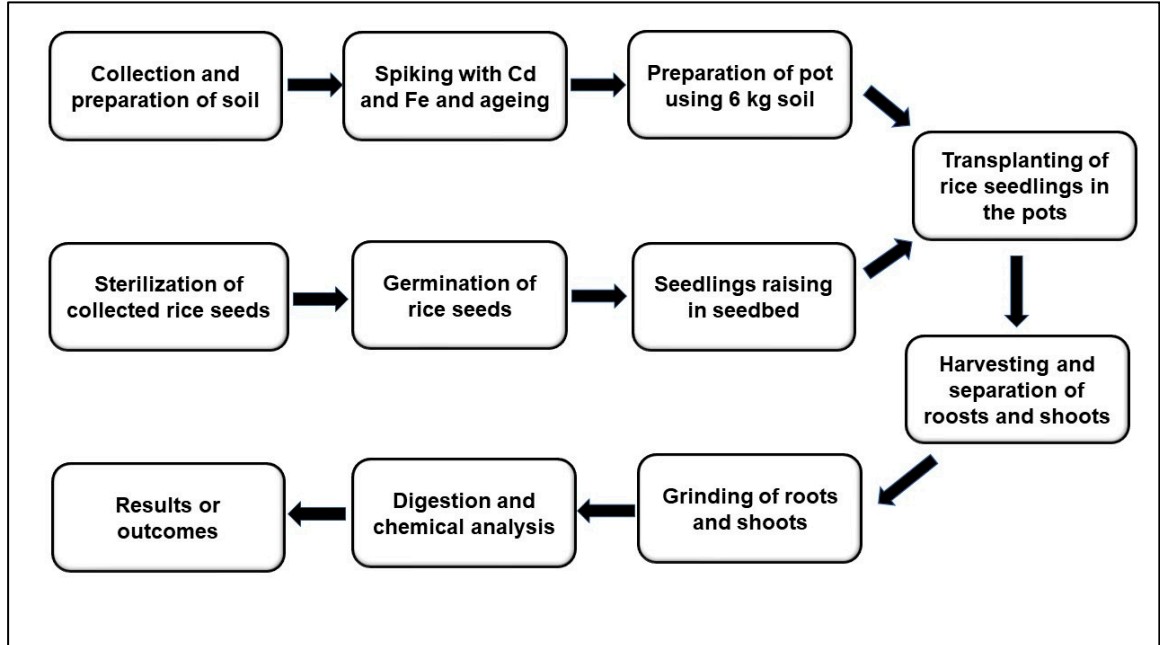

**Scheme 1.** Flow diagram showing the entire experimental procedure.

### 2.3. Seedlings Establishment and Fertilizer Application

Rice seeds were disinfected in a solution of 1% (v/v) sodium hypochlorite (NaOCl) with one drop of Decon for 30 min, followed by thorough washing with deionized water and then Milli-Q water [30]. The seeds were placed in plastic tray and covered with wet paper towel for germination at a constant temperature room (30 °C) for about 3 days. The germinated rice seeds were then transferred to seedling-raising media on 31 March 2019 and grown for 22 days. Each pot was supplied with urea, TSP, muriate of potash, and gypsum at a rate of 200 mg N $kg^{-1}$ soil, 65 mg P $kg^{-1}$ soil, 160 mg K $kg^{-1}$ soil, and 75 mg S $kg^{-1}$ soil, respectively. Two days before transplanting rice seedlings, the full doses of P, K, and S fertilizer except N fertilizer were properly mixed with potted soils. Nitrogen fertilizer was supplied to pot soil in 3 equal splits at 7, 30, and 60 days after transplanting. Twenty-five-day-old uniform and healthy seedlings were carefully uprooted and transplanted into each pot. Two hills were placed in each pot maintaining equal distance, and each hill contained two rice seedlings. Intercultural operations such as weeding, irrigation, and soil loosening were done as per the requirement to provide the favourable conditions of plant growth and development.

### 2.4. Experimental Treatments

Before transplanting rice seedlings, the potted soils were spiked with three levels of Fe and Cd and then aged for 30 days. The Fe was supplied as $FeSO_4.7H_2O$ at a level of 0 (Fe0), 1.0 (Fe1.0), and 2.0 (Fe2.0) g Fe $kg^{-1}$ soil, and Cd was added as $Cd(NO_3)_2.4H_2O$ at a level of 0 (Cd0), 1 (Cd1.0), and 3 (Cd3.0) mg Cd $kg^{-1}$ soil. The levels of Fe and Cd were selected based on the literature [13,28]. In Japan, the Cd in soil ranged from 0.144 to 3.76 mg $kg^{-1}$ [31]. Both Cd and Fe were supplied in the solution form and mixed thoroughly with the potting soil. The Fe was supplied to stimulate iron plaque around the root to varying amounts, and Cd addition was done to represent Cd contaminated soil. The Fe and Cd combination generated nine treatments in total: Fe0Cd0, Fe0Cd1.0, Fe0Cd3.0, Fe1.0Cd0, Fe1.0Cd1.0, Fe1.0Cd3.0, Fe2.0Cd0, Fe2.0Cd1.0, and Fe2.0Cd3.0. Potted soil was maintained under flooded conditions with a layer of water of about 2–3 cm above the soil surface during the whole period of plant growth experiment.

### 2.5. Harvesting and CBD Extraction of Iron Plaque from Roots

Rice plants were harvested at 7 weeks of transplanting and carefully washed with tap water followed by ultra-pure water for approximately 3–5 min to remove excess soil. The plants were then cut into roots and shoots portion with sharp stainless-steel scissors. To measure the Fe and Cd concentrations in iron plaque, citrate-bicarbonate-dithionite (CBD) method was deployed to quantify the iron plaque on the surface of fresh rice root [32,33]. Briefly, the whole rice roots from every plastic pot were firstly incubated for 70 min at room temperature (25 °C) in 80 mL 0.03 M sodium citrate ($Na_3C_6H_5O_7.7H_2O$), 0.125 M sodium bicarbonate ($NaHCO_3$), and 1.6 g sodium dithionite ($Na_2S_2O_4$). Roots were washed three times with Milli-Q water, and the solution was combined with the 80-mL CBD extracts. The final solution was then shifted into 100 mL volumetric flasks and filtered with syringe filter (Minisart CA 0.45 um 28 mm) into plastic containers. The extracted solution was stored at 4 °C for further analysis. After CBD extraction, the roots and shoots were transferred to a drying oven at 70 °C for 3 days to achieve constant weight. Then, the root and shoot biomass were recorded according to the dry weight by the use of analytical balance.

### 2.6. Digestion and Analysis of Rice Plant Parts

Prior to analysis, the oven-dried rice plant samples were ground to powder using a stainless-steel grinder. Plant samples were digested and analysed for Fe and Cd content according to the methods described by [34]. After digestion, Fe concentration in digested solutions and CBD extracts were determined by a dual view (Axial and radial) inductively coupled plasma optical emission spectrometry (ICP-OES, PerkinElmer Avio 200, USA) and Cd concentrations were measured by inductively coupled plasma mass spectrometry

(ICP-MS, Agilent 7900, Japan) coupled with an autosampler (Agilent Technologies, SPS 4). In order to monitor the quality of instrumental analysis and to examine the accuracy of results, quality control measures, which included reagent blank, duplicate samples, continuing check verification (CCV), and standard reference material (SRM), were taken consistently during the course of pre-treatment and analysis.

### 2.7. Analytical Performances

A standard reference plant material such as rice flour (SRM 1568 b, National Institute of Standards and Technology (NIST), Gaithersburg, Maryland, USA) was included to verify the accuracy and precision of the digestion procedure and subsequent analysis. The certified value of Fe and Cd were $7.42 \pm 0.44$ µg kg$^{-1}$ and $22.4 \pm 1.3$ µg kg$^{-1}$, whereas the observed value were $6.84 \pm 0.25$ µg kg$^{-1}$ and $20.18 \pm 2.6$ µg kg$^{-1}$, indicating the recovery values of 92% and 90%, respectively.

### 2.8. Statistics Used

All experimental data were evaluated by using statistical software JMP version 14.2.0. Three-way full factorial ANOVA was applied to detect the significant effect of Fe and Cd addition on plant biomass and Fe and Cd concentration in CBD extracts and in the rice seedlings. Other measurable investigation was moreover conducted by utilizing Microsoft Excel 2016. Probability level of 0.05 or 0.01 was considered to be statistically significant. Means and standard error (SE) were estimated from the fitted model and means were compared according to the Student's *t* test. GraphPad Prism software version 8.4.3 was utilized to produce all figures. The values for all parameters were expressed as means $\pm$ standard error (SE).

## 3. Results

### 3.1. Biomass of Rice Seedlings

The biomass of the shoot and roots (dry weight) of Langi and Quest rice cultivars grown in red and grey soil with various levels of Fe and Cd is shown in Figure 1. The shoot and roots biomass of the Quest cultivar was significantly higher ($p < 0.001$) than that of the Langi cultivar in both soil types. Relative to Cd control, the shoot biomass of the Quest cultivar decreased by 12.6% and 23.2% in red soil and 12.2% and 18.1% in grey soil in the Cd1.0 and Cd3.0 treatments, whereas the shoot biomass of the Langi cultivar was decreased by 15.9% and 30.8% in red soil and 12.8% and 19.4% in grey soil in the Cd1.0 and Cd3.0 treatments, respectively. Compared with the Cd control, the root biomass decrement was 8.6% and 16.4% in red soil and 8.8% and 16.0% in grey soil for the Quest cultivar and 9.7% and 18.8% in red soil and 11.2% and 19.2% in grey soil for the Langi cultivar in Cd1.0 and Cd3.0, respectively. The results indicated that the toxic effect of Cd on the shoots and roots biomass was less in the Quest cultivar compared to the Langi cultivar.

The shoot biomass increment was 11.2% and 19.5% in red soil and 7.4% and 19.4% in grey soil for the Quest cultivar and 26.0% and 43.3% in red soil and 28.1% and 43.1% in grey soil for the Langi cultivar in the Fe1.0 and Fe2.0 treatments as compared with Fe0. For the Quest cultivar, the root biomass increased by 11.1% and 20.3% in red soil and 10.4% and 19.3% in grey soil, while the root biomass of the Langi cultivar was increased by 13.3% and 22.9% in red soil and 19.0% and 27.4% in grey soil in Fe1.0 and Fe2.0 as compared to the Fe control. These results suggested that the shoot and roots biomass of rice seedlings were significantly increased with Fe supply to the soil. However, there was a significant difference ($p < 0.001$) in biomass of shoots and roots between the cultivar with the ranking of Quest > Langi. Our results also indicated that Cd application potentially retards the biomass production of Langi cultivar, while Fe supply mitigates the toxic effects induced by Cd in this rice cultivar.

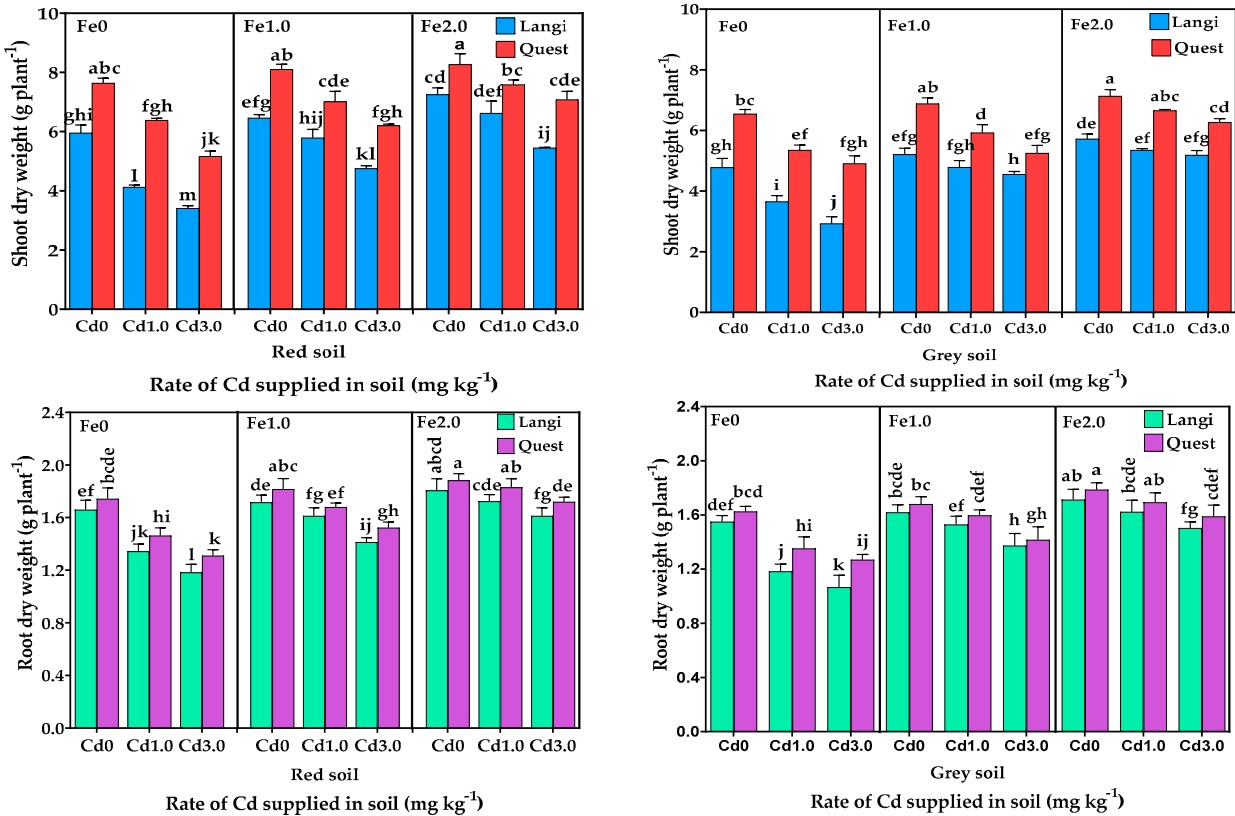

**Figure 1.** Effect of different levels of Fe (Fe0, Fe1.0, and Fe2.0: 0, 1.0, and 2.0 g Fe kg$^{-1}$ soil) and Cd (Cd0, Cd1.0, and Cd3.0: 0, 1.0, and 3.0 mg Cd kg$^{-1}$ soil) on the biomass of shoots and roots of two rice cultivars grown in different soil. Mean ($\pm$SE) were calculated from three replicates for each treatment. Bars with different letters are statistically significant at $p \leq 0.001$ applying Student's *t* test.

### 3.2. Concentration of Fe and Cd in Iron Plaque on the Root Surface of Rice Cultivars

A visible reddish-brown coating appeared around the roots of the rice cultivar. The concentrations of Fe and Cd in iron plaque extracted from the root surface of rice cultivar are shown in Figures 2 and 3. The concentration of CBD-Fe was enhanced with the escalating Fe application in soil, and the highest value was observed in Fe2.0 than in Fe1.0 or Fe0. In both red and grey soils, the application of 1.0 and 2.0 g kg$^{-1}$ Fe significantly increased the concentration of Fe in the CBD extracts of both rice cultivars ($p < 0.001$) compared with the Fe control treatment (Fe0), and the difference between Fe1.0 and Fe2.0 were also significant ($p < 0.001$). For the Quest cultivar, the highest CBD Fe was observed in Cd0 but was significantly different from the other two Cd levels in Fe1.0 ($p < 0.0001$) in both red and grey soils. Similarly, the highest value of CBD-Fe was also observed in Cd0, which was statistically different from Cd1.0 and Cd3.0 in Fe2.0. For the Langi cultivar, the CBD-Fe concentrations were not significant at two Cd supply with both Fe1.0 and Fe2.0 treatments in both soil types. Our results also indicated that the Quest cultivar had significantly ($p < 0.001$) higher Fe concentration in CBD extracts than that of the Langi cultivar in both red and grey soils, regardless of Cd supply. The CBD-Fe concentration between the Quest and Langi rice cultivars was not significant in Fe0 with the addition of Cd in both soil types.

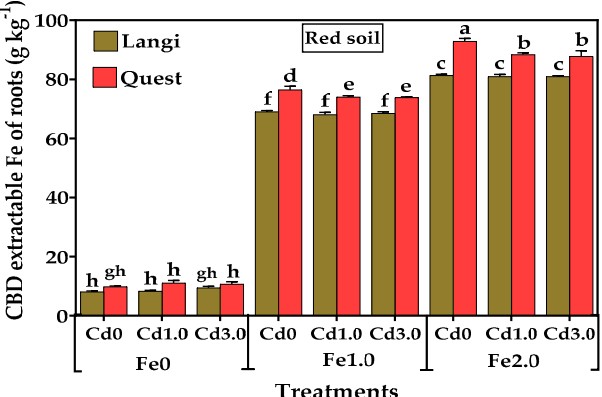 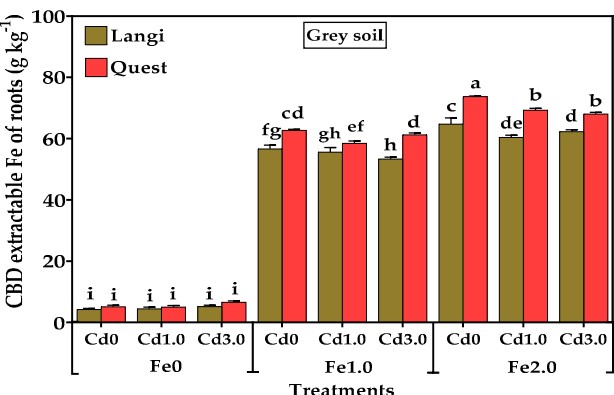

**Figure 2.** Effect of different levels of Fe (Fe0, Fe1.0, and Fe2.0: 0, 1.0, and 2.0 g Fe kg$^{-1}$ soil) and Cd (Cd0, Cd1.0, and Cd3.0: 0, 1.0, and 3.0 mg Cd kg$^{-1}$ soil) on CBD-extractable Fe on roots of two rice cultivar grown in different soil. Mean ($\pm$SE) were calculated from three replicates for each treatment. Bars with different letters are statistically significant at $p \leq 0.001$ applying Student's $t$ test.

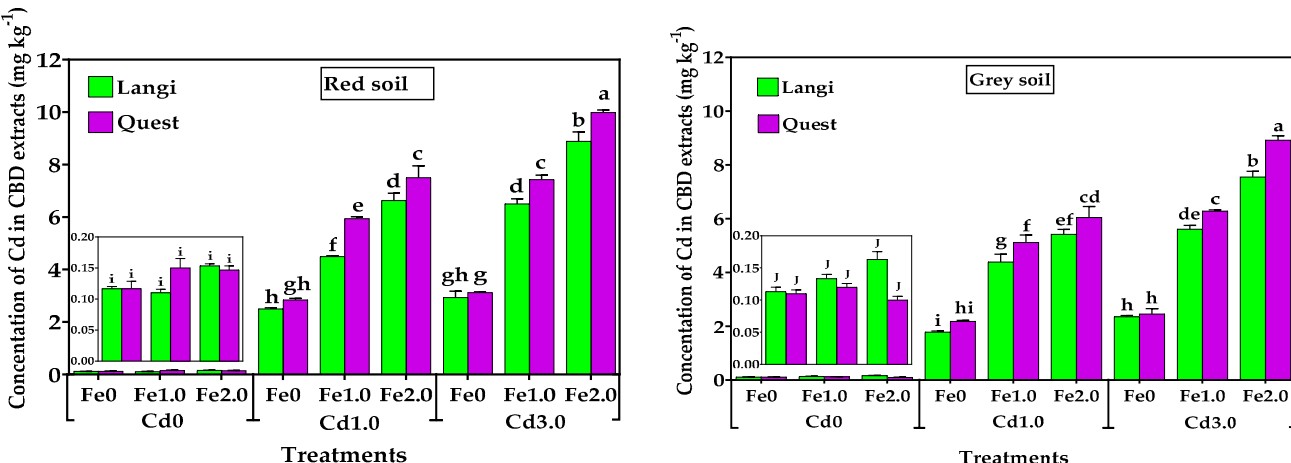

**Figure 3.** Effect of different levels of Fe (Fe0, Fe1.0, and Fe2.0: 0, 1.0, and 2.0 g Fe kg$^{-1}$ soil) and Cd (Cd0, Cd1.0, and Cd3.0: 0, 1.0, and 3.0 mg Cd kg$^{-1}$ soil) on Cd concentration in CBD extracts on root surface of two rice cultivar grown in different soil. Mean ($\pm$SE) were calculated from three replicates for each treatment. Bars with different letters are statistically significant at $p \leq 0.001$ applying Student's $t$ test.

The Cd concentration in CBD extracts was significantly affected by both Fe and Cd addition in both soils at the seedling stage (Figure 3). The Fe supply significantly ($p < 0.001$) increased the concentration of Cd in CBD extracts except at Cd0 (Figure 4). There was no significant difference between two rice genotypes in Cd0 with the addition of Fe in both soils. Relative to the Fe control, the Cd concentration in CBD extracts of the Quest cultivar increased by 109% and 166% in red soil and 146% and 204% in grey soil in the Fe1.0 and Fe2.0 treatments, whereas the Cd concentration in CBD extracts of the Langi cultivar increased by 80% and 164% in red soil and 134% and 177% in grey soil in the Fe1.0 and Fe2.0 treatments, respectively, in the presence of Cd1.0.

Similarly, in the Cd3.0 treatment, the CBD-Cd concentrations of the Quest cultivar rose by 138% and 222% in red soil and 156% and 263% in grey soil in the Fe1.0 and Fe2.0 treatments, respectively, whereas the CBD-Cd concentrations' increment in the Langi cultivar were 122% and 203% in red soil and 136% and 220% in grey soil in the Fe1.0 and Fe2.0 treatments, respectively, compared with the Fe0. Our results demonstrated that Cd concentration in CBD extracts increased with increasing Fe supply and was influenced by the rice cultivar, soil type, and external Cd concentration. In addition, CBD Cd concentrations for both soil types were positively correlated with CBD Fe concentrations (Figure 4).

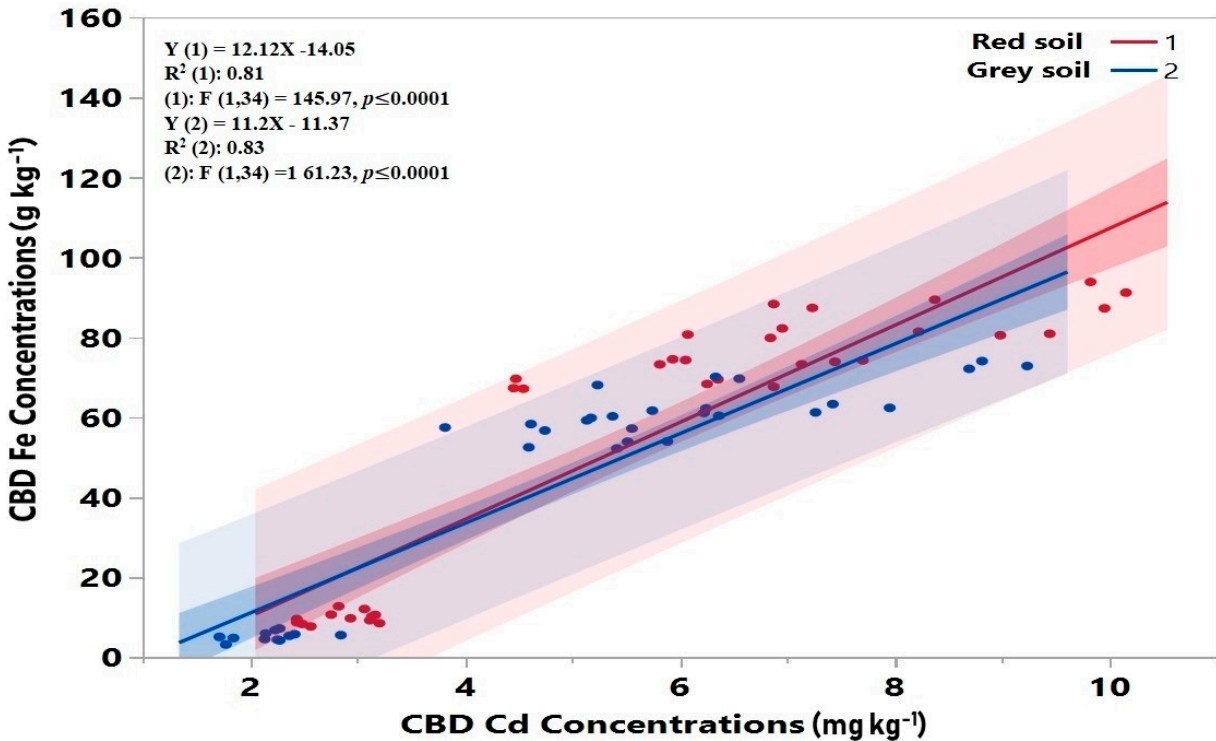

**Figure 4.** Relationship between concentration of Fe and Cd in CBD extracts. Linear regression model was checked by linearity and $R^2$ value (correlation coefficient).

### 3.3. Concentration of Fe in Rice Seedlings

The concentrations of Fe in shoots and roots of rice seedlings are shown in Figure 5. The shoots' Fe levels were significantly influenced by rice cultivar, Fe treatment, and soil type. In both the red and grey soils, the addition of 1.0 and 2.0 g kg$^{-1}$ Fe significantly increased Fe concentration in the shoot and roots of the rice cultivars as compared to the control (Fe0), and the difference between the Fe1.0 and Fe2.0 treatments were also statistically significant ($p < 0.001$). Relative to the control, the application of 1.0 g kg$^{-1}$ increased the shoot Fe by 70%, while the addition of 2.0 g kg$^{-1}$ Fe increased it by 96% in red soil. For grey soil amended with 1.0 and 2.0 g kg$^{-1}$ Fe, the shoot Fe levels were 86% and 115% higher, respectively, than those in the control seedling shoot. Under the soil Fe treatments of 1.0 and 2.0 g kg$^{-1}$, the Fe concentration in the shoots of the Quest cultivar were higher than that of the Langi cultivar in Cd1.0 and Cd3.0, and this difference in shoot Fe concentrations between the two rice cultivars was statistically significant in both red and grey soils. For both red and grey soils, there was no significant difference observed between the two rice cultivars in both Cd supply levels when no Fe was supplied in the soil. The concentrations of Fe in the roots of both rice cultivars were higher than the shoots in all treatments (Figure 5). Relative to the control, the root Fe levels were 5.4- and 6.5-fold higher under the application of 1.0 and 2.0 g kg$^{-1}$ Fe, respectively, in red soil. For grey soil amended with 1.0 and 2.0 g kg$^{-1}$ Fe, the root Fe concentrations were 4.1 and 5.8 times higher than control, respectively. In red soil, the application of 2.0 g kg$^{-1}$ Fe significantly increased the Fe concentration in the roots of the Quest cultivar compared to the Fe1.0 treatment in both Cd supply levels, and there was no significant difference observed between Fe1.0 and Fe2.0 in the root Fe concentrations of the Langi cultivar in both Cd supply levels. For both rice cultivars, the root Fe concentration was significantly higher in Fe2.0 than in Fe1.0, but there was no significant difference observed between the two rice genotypes in the Cd1.0 and Cd3.0 treatments under both Fe supply levels. For both red and grey soil, there was no significant difference observed between the two rice cultivars in both Cd supply levels when no Fe was supplied in the soil.

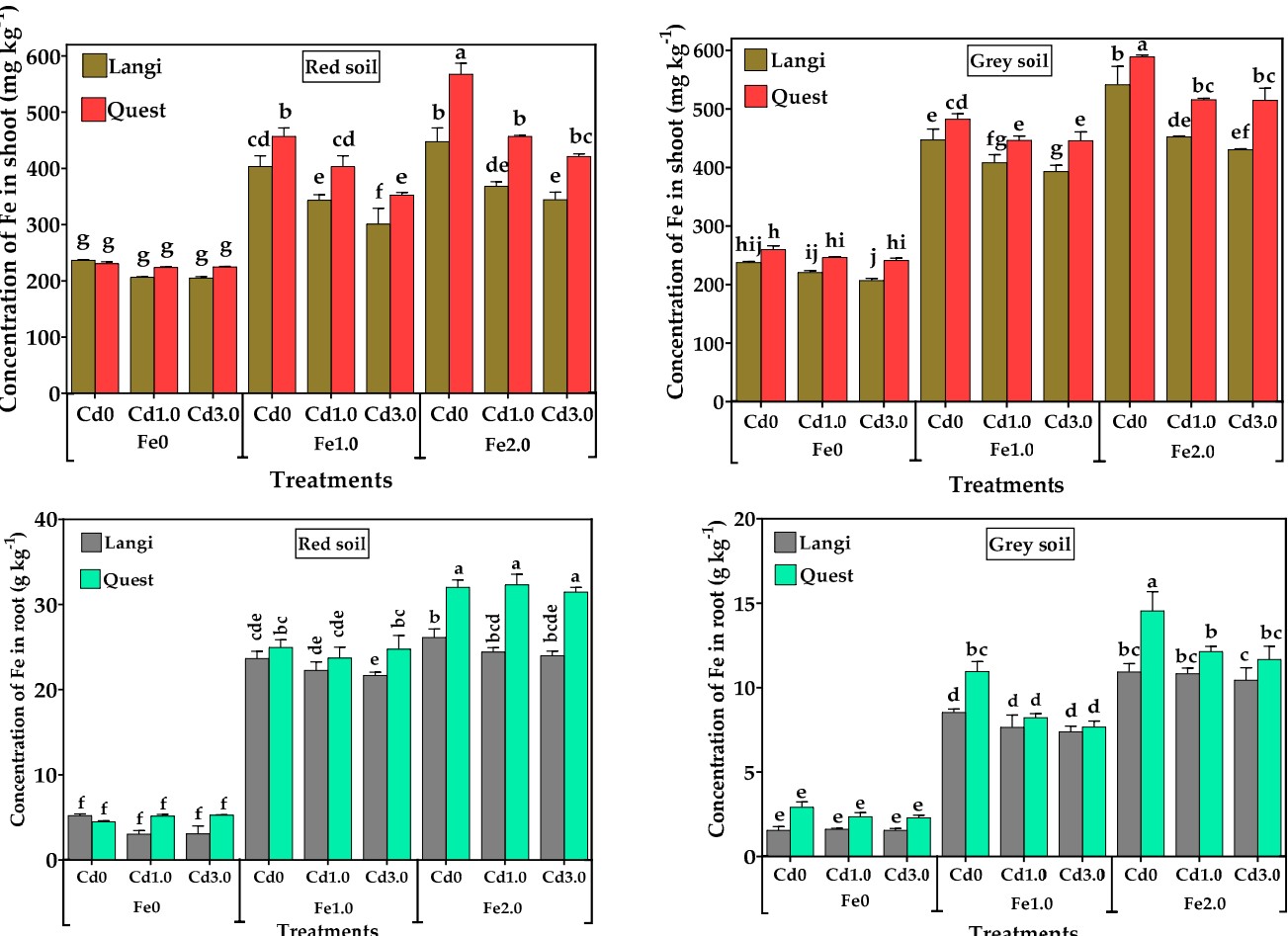

**Figure 5.** Effect of different levels of Fe (Fe0, Fe1.0, and Fe2.0: 0, 1.0, and 2.0 g Fe kg$^{-1}$ soil) and Cd (Cd0, Cd1.0, and Cd3.0: 0, 1.0, and 3.0 mg Cd kg$^{-1}$ soil) on Fe concentration in the shoots and roots two rice cultivars grown in different soil. Mean ($\pm$SE) were calculated from three replicates for each treatment. Bars with different letters are statistically significant at $p \leq 0.001$ applying Student's $t$ test.

### 3.4. Concentration of Cd in Rice Seedlings

There were wide variations between the two rice genotypes in the Cd concentration of rice shoots and roots for the two soil types (Figure 6). The presence of Cd in the shoot and root of rice genotypes with no Cd supplementation to the soil is likely to be from the background soil Cd. In general, the concentration of Cd in rice shoots and roots was boosted with the rising Cd levels and decreased with increasing Fe levels. For the Langi cultivar, the Cd concentrations in shoots were 38.2- and 56.1-fold higher in Fe0, 25.3- and 38.9-folds higher in Fe1.0, and 21.7- and 30-fold higher in Fe2.0 with Cd1.0 and Cd3.0 treatments, respectively, compared to the corresponding Cd control in red soil. For the Quest cultivar, the Cd concentrations in shoots were 33.7- and 49.2-fold higher in Fe0, 20.4- and 30.1-fold higher in Fe1.0, and 15.5- and 22.2-fold higher in Fe2.0 with the addition of 1.0 and 3.0 mg kg$^{-1}$ Cd, respectively, compared to corresponding Cd control in red soil. For the Langi cultivar, the Cd concentrations in shoots were 12.8- and 17.1-fold higher in Fe0, 7.9- and 8.4-fold higher in Fe1.0, and 6.3- and 8.2-fold higher in Fe2.0 under 1.0 and 3.0 mg kg$^{-1}$ Cd supplementation, respectively, compared to corresponding Cd control in grey soil. For the Quest cultivar, the Cd concentrations in shoots were 12.3- and 15.9-fold higher in Fe0, 6.7- and 7.7-fold higher in Fe1.0, and 5.7- and 7.8-fold higher in Fe2.0 with Cd1.0 and Cd3.0 supplementation, respectively, compared to the corresponding Cd control in grey soil.

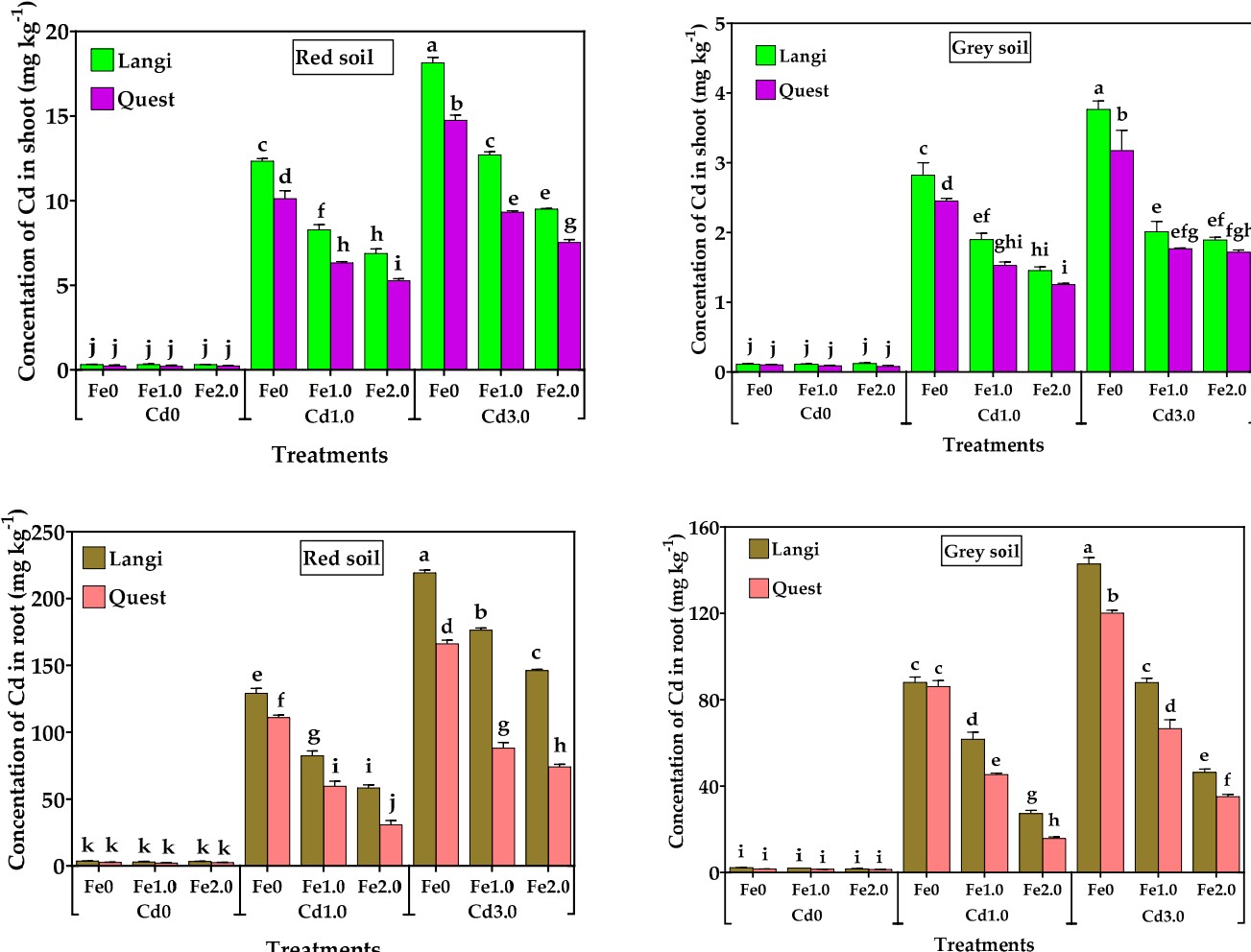

**Figure 6.** Effect of different levels of Fe (Fe0, Fe1.0, and Fe2.0: 0, 1.0, and 2.0 g Fe kg$^{-1}$ soil) and Cd (Cd0, Cd1.0, and Cd3.0: 0, 1.0, and 3.0 mg Cd kg$^{-1}$ soil) on Cd concentration in shoots and roots of two rice cultivars grown in different soil. Mean (±SE) were calculated from three replicates for each treatment. Bars with different letters are statistically significant at $p \leq 0.001$ applying Student's $t$ test.

In red soil, relative to the corresponding Cd control, the Cd concentrations in the roots of the Langi cultivar were 49.7- and 84.3-fold higher in Fe0, 35.9- and 76.7-fold higher in Fe1.0, and 29.3- and 73.1-fold higher in Fe2.0, whereas the Cd concentrations in the roots of the Quest cultivar were 44.4- and 66.5-fold higher in Fe0, 29.9- and 44.1-fold higher in Fe1.0, and 17.6- and 42.4-fold higher in Fe2.0 with 1.0 and 3.0 mg kg$^{-1}$ Cd addition, respectively. In grey soil, relative to the corresponding Cd control, the Cd concentrations in the roots of the Langi cultivar were 38.7- and 62.8-fold higher in Fe0, 41.2- and 58.7-fold higher in Fe1.0, and 17.6- and 42.4-fold higher in Fe2.0, whereas the Cd concentrations in the roots of the Quest cultivar were 38.3- and 53.4-fold higher in Fe0, 30.7- and 45.1-fold higher in Fe1.0, and 11.9- and 22.0-fold higher in Fe2.0 with 1.0 and 3.0 mg kg$^{-1}$ Cd addition, respectively.

## 4. Discussion

One of the most serious consequences of Cd toxicity is the suppression of plant growth and development, which leads to a reduction in biomass production [17]. Cd stress has been shown to impair rice growth and biomass in numerous studies [35–37]. The shoot and root biomass of Quest and Langi cultivars were observed to decrease with the addition of Cd to the soil in this study, but the toxic effect of Cd on shoots and roots biomass was smaller in Quest than in Langi. Our results also indicated that the shoot and root biomass reduction was more pronounced under high Cd stress (Cd3.0) in both soil types. Our

findings were in line with Feng et al. (2017) who showed that shoot and root dry weight showed remarkable reduction after Cd exposure, and the alteration was more obvious under high Cd stress (5 mg kg$^{-1}$) [38]. Plant height, biomass, and photosynthesis were reduced, whereas oxidative stress increased under Cd stress in rice [39]. The reduction in photosynthesis might be responsible for the reduced biomass production of rice under Cd-stressful conditions [39,40]. In the present experiment, the shoot and root biomass of the two rice cultivars significantly increased with the Fe supply levels in both soils, with the biomass increment being greater in the Quest cultivar than in the Langi cultivar. This observation was strongly supported by a recent investigation by Siddique et al. (2021) who reported that rice biomass in terms of straw and grain yield significantly increased with Fe supplementation [41]. Similar findings were reported by Liu et al. (2017), who attributed the varying biomass pattern for the two rice types (Shennong 265 and Shendao 6) is due to the differing reactions to Fe and (or) Cd treatments. Shennong 265's growth was aided by Fe and not hampered by Cd, but Shendao 6's growth was hampered by Cd and marginally aided by Fe [17]. When no Fe was added, both soil Cd1.0 and Cd3.0 significantly decreased the shoot and root biomass of the Quest and Langi cultivars (Figure 1). The addition of 1.0 or 2.0 g kg$^{-1}$ Fe significantly increased the biomass of the two rice cultivars. The findings showed that Fe supplementation had the largest impact on rice growth and that adding Fe to the soil reduces the negative influence of Cd on rice seedlings' physiological parameters to some extent.

The present study showed that iron plaque formation increased when the soil was supplemented with extra Fe (Figure 2). We also found that the Quest cultivar had a higher ability to the formation of iron plaque than the Langi cultivar. Because iron plaque has a high affinity for metal (loids), it impacts the plant uptake of these elements. There are numerous reports that show a significant positive relationship between metal(loids) and Fe concentration in iron plaque [27,29,42–45]. A significant positive association was observed between Fe and Cd in iron plaque (Figure 4), as well as an increased Cd concentration in iron plaque with increasing Fe supply, suggested that iron plaque had strong Cd adsorption ability, reducing Cd translocation in rice plants. Several investigations have discovered that iron plaques contain both crystalline and amorphous iron oxides and that because these iron oxides share features, iron plaque has a high potential for heavy metal adsorption [46–48]. Iron plaque has been shown in numerous studies to immobilize Zn, Cd, As, Hg, and Cr in the environment, reducing their uptake by plants [27,42,45,49,50], but other researchers showed the opposite trend [51,52]. Several factors, including soil pH, Eh, the quantity of iron plaque, metal ion type and concentration, and plant species and kinds, may be to blame for the inconsistency. We observed that 1.0 and 2 g kg$^{-1}$ Fe effectively reduce Cd concentrations in rice seedling shoots and roots, with the rate of decline being faster in Fe2.0 than Fe1.0 when compared to the Fe control. Zhang et al. (1998) found that iron plaque with 50 mg L$^{-1}$ Fe lowered Zn content in shoots [49], which is consistent with our findings. Liu et al. (2007) found that iron plaque considerably reduced the concentration of Cd in the shoots of rice seedlings when a high dose of Fe (100 mg L$^{-1}$ Fe) was applied [52]. Furthermore, Liu et al. (2008) found that external Fe supplementation (1 or 2 g kg$^{-1}$ Fe) significantly reduced the concentration of Cd in the shoots and roots of rice seedlings growing in soil, regardless of whether the Cd content was low or high (2 or 10 mg kg$^{-1}$ Cd) [27]. Similar results were also obtained by Sebastian and Prasad (2016), who reported that Cd concentration in rice plant decreased with exogenous application of Fe in soil [28]. In addition, to restrict Cd accumulation in plants, Fe amendment demolished ROS, minimized oxidative stress, and enhanced plant growth [53]. Cd is a redox inactive heavy metal that generates ROS through indirect processes such as by triggering the Nicotinamide adenine dinucleotide phosphate (NADPH) oxidase function in the plant tissue, which eventually results in oxidative stress in plant tissues [53]. Several studies reported that Cd had significant impact on ROS production in various plant species [53,54]. The Cd stress increased the ROS levels, including hydrogen peroxide and superoxide, which enhance the lipid peroxidation rate and ultimately destroy the antioxidant defense metabolism in

plant cells in the absence of Fe. Significant increases in malondialdehyde (MDA) levels were also observed in plants [53]. So, the addition of Fe supplements may reduce the MDA levels. Thus, in this present study, Fe application (1 and 2 g kg$^{-1}$) significantly reduced Cd accumulation in plant resulted in increased biomass production due to improvement of antioxidant defense mechanism through scavenging ROS metabolism.

Our results also demonstrated that the Quest cultivar accumulates low levels of Cd while the Langi cultivar accumulates higher levels of Cd in both soil types. Cd absorption and transport in crops differs significantly between plant species and cultivars within the same species [55–57]. The Cd accumulation and dispersion in various parts of the rice cultivar are caused by dominant heritable traits [58,59]. In 49 rice cultivars grown on polluted soil, a considerable fluctuation of a roughly 13–23-fold difference in Cd levels was discovered [60,61]. Liu et al. (2007) found a wide range of Cd concentrations in polished rice, ranging from 0.14 to 1.43 mg kg$^{-1}$ [62]. Cd concentrations in brown rice ranged from 0.96 to 0.99 mg kg$^{-1}$ in 38 rice genotypes [63]. Ye et al. (2012) investigated the effect of rice genotypes on Cd concentration in various organs of rice plants and found that Cd levels of various parts of indica rice varieties were higher than that in hybrid and japonica cultivars [64]. In a field setting, Römkens et al. (2011) found that indica cultivars had increased Cd absorption in low Cd soil and that only unpolluted soils were suitable for indica cultivars [65]. A pot experiment was conducted by Song et al. (2015) in glasshouse conditions, and it was observed that there existed a significant difference in the Cd levels in various plant parts of rice (roots and straw) in 20 rice cultivars under the same soil Cd levels (0.3 or 0.6 mg kg$^{-1}$) [10]. Similarly, late rice varieties showed remarkably higher content of Cd than that in early rice varieties in both grains and rachises when planted at the same site [66]. Hongjiang et al. (2013) found that Cd resistance and build-up differed considerably across 146 rice lines in hydroponic trials and 17 rice lines in the pot culture [67]. The ability of different rice cultivars to absorb and accumulate Cd was linked to their root oxidation abilities, root acidity, and root organic acid secretions [64]. Therefore, rice cultivars with higher Cd accumulation tendencies should be avoided in the agricultural production system as much as possible.

## 5. Conclusions

Our present study demonstrated that the Langi rice cultivar is more susceptible to exogenous Cd application than the Quest rice cultivar. The application of Fe significantly increased the biomass production in the Quest cultivar compared to the Langi rice cultivar in the absence or presence of Cd addition into the soil. The higher Fe and Cd concentrations were observed in the iron plaque extracted from the roots of the Quest cultivar than the Langi cultivar in the Fe1.0 and Fe2.0 treatments. The Fe concentration in the roots and shoots of the Quest cultivar was higher than the Langi cultivar and Fe concentration in rice tissue were increased with Fe supplementation under the influence of Cd application in soil. The Cd concentration in the roots and shoots of Langi genotype was more than the Quest cultivar and Cd concentration decreased with increasing Fe levels in soil. This study results revealed that the presence of more Cd in iron plaque might be responsible for lower Cd accumulation and transfer to above-ground parts of rice plants and root plaque formation serves as a source of Fe that prevented the Cd toxicity and Cd-induced Fe deficiency. Our results also indicated that the soil type significantly affected the biomass production, iron plaque formation, and the Fe and Cd concentration in Quest and Langi rice cultivars. The results of this present study will provide useful information for the better management of Cd-contaminated soil and for increasing food safety, quality, and stability, as well as the economic benefit of minimizing health risks for populations relying on rice-based diets. However, the characterization of Cd allocation in the root iron plaque of rice seedlings and the transfer and remobilization mechanisms of Cd from the root to shoot of the rice plant with a view to reducing Cd in edible parts requires further investigation.

**Author Contributions:** A.B.S.: conceptualization, designed and conducted the experiment, writing—original draft, formal analysis, statistical analysis, visualization, data collection; M.M.R.: project

administrator, conceptualization, supervision, formal analysis, writing—review and ed-iting; M.R.I.: methodology, formal analysis, writing—review and editing; M.T.S.: analysis, writ-ing—review and editing; B.N.: writing—review and editing; R.N.: supervision, writing—review and editing. All authors have read and agreed to the published version of the manuscript.

**Funding:** This research received no external funding.

**Institutional Review Board Statement:** Not applicable.

**Informed Consent Statement:** Not applicable.

**Data Availability Statement:** The raw data that supported this research will be made available upon reasonable request to the authors without undue reservation.

**Acknowledgments:** The first author is pleased to acknowledge the scholarship support from the University of Newcastle and the Australian Government's Research Training Program. The present research work also received financial and infrastructural support from the Cooperative Research Centre for Contamination Assessment and Remediation of the Environment (CRC-CARE). We also express sincere thanks to Kim Colyvas, Manager Statistical Consulting Unit, School of Mathematical and Physical Sciences, the University of Newcastle, for providing statistical support. The Australian Rice Partnership that includes Sunrice, AgriFutures, and NSW DPI are acknowledged for the generous gift of rice germplasm to use in this study. NSW DPI provided the rice seed under the MTA, which is covered by the Australian Rice Partnership.

**Conflicts of Interest:** The authors declare no conflict of interest.

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
