# Peer review of "Influence of Iron Plaque on Accumulation and Translocation of Cadmium by Rice Seedlings"

_sustainability, doi:10.3390/su131810307_

Round 1
Reviewer 1 Report
Dear Authors,
As per my view, paper is interesting and falls within the scope of the journal. Authors try to see the influence of iron plaque on accumulation and translocation of cadmium by rice seedlings: variation of soil type and rice cultivar. These results indicate that enhanced iron plaque formation on the root surface was crucial to reduce Cd concentration in rice plants; which could be an effective strategy to regulate grain Cd accumulation of rice plants. Further, langi rice cultivar was more susceptible to exogenous Cd application than the Quest rice cultivar. -
Minor comments:
Authors need to prepared the article as per author guidelines (journal format)
Sub-script and super-script are not been taking care, kindly check and correct thoroughly (e.g., Fe2+, Fe3+, 5Fe2O3.9H2O etc.)
Line no 46-49: need support (see suggested recent publications; https://doi.org/10.1016/j.chemosphere.2020.128855; J Agric Food Chem. 68(47):13497-13529. doi: 10.1021/acs.jafc.0c04579).
Kindly provide a flow diagram to understand the methodologies used in this study in clear way
Section 2.1 need to be briefly explain (line no 113-117 need to be shift to results and discussion section)
Heading of Section 2.8 should be statistics used
Author Response
Reviewer 1
Dear Authors, As per my view, paper is interesting and falls within the scope of the journal. Authors try to see the influence of iron plaque on accumulation and translocation of cadmium by rice seedlings: variation of soil type and rice cultivar. These results indicate that enhanced iron plaque formation on the root surface was crucial to reduce Cd concentration in rice plants; which could be an effective strategy to regulate grain Cd accumulation of rice plants. Further, langi rice cultivar was more susceptible to exogenous Cd application than the Quest rice cultivar. Therefore, paper having enough novelties and could be accepted for publication after minor suggestion suggested below:-
Response: We appreciate reviewer for the excellent comment and strong recommendation to publish our article.
Minor comments:
Comment: Authors need to prepare the article as per author guidelines (journal format)
Response: We have formatted the manuscript as per journal guidelines.
Comment: Sub-script and super-script are not been taking care, kindly check and correct thoroughly (e.g., Fe2+, Fe3+, 5Fe2O3.9H2O etc.)
Response: We are sorry for that. We have now rechecked and corrected all sub-scripts and super-scripts throughout the revised manuscript.
Comment: Line no 46-49: need support (see suggested recent publications; https://doi.org/10.1016/j.chemosphere.2020.128855; J Agric Food Chem. 68(47):13497-13529. doi: 10.1021/acs.jafc.0c04579).
Response: We have included references suggested by the reviewer (Please see lines 52-54, track change mode).
Comment: Kindly provide a flow diagram to understand the methodologies used in this study in clear way.
Response: We have provided a flow diagram of the methodologies (Scheme 1) used in the experiment in the revised manuscript as per reviewer suggestion (please see line 145, page 7, track change mode).
Comment: Section 2.1 need to be briefly explain (line no 113-117 need to be shift to results and discussion section)
Response: We are bit confused about this comment as we think lines no 113-117 (lines 127-129 in revised manuscript) are more appropriate in the Method section rather than results and discussion section. Hence, we have kept the text as it is in the revised manuscript.
Comment: Heading of Section 2.8 should be statistics used
Response: Done in the revised manuscript (please line 209).
Reviewer 2 Report
REVIEW REPORT
TITLE: Influence of iron plaque on accumulation and translocation of cadmium by rice seedlings: variation of soil type and rice cultivar
## Title change suggested. Title has to be concise yet attractive.
# Authors sequence need to be addressed properly.
Abstract:
Line No.20: “influence of Fe addition” amendment / supplementation is more appropriate.
In general abstract is considered as a gist of the entire article in a very precise manner. A good abstract should have the experimental design and some of the important data or statistical findings.
## Rewriting suggested.
Keywords: Ok
Introduction:
Line No. 66: “Unlike other aquatic plants” Justify and/ change suggested.
While writing an article on metallic toxicant, it is expected to have permissible limits and recent evidence on similar interaction study in the introduction section.
- Materials and methods
2.1. Soil collection, processing and preparation
## which soil sapling method did the authors actually followed?
GCER-full name
## Capacity of pots?
## Is there any particular reason behind choosing these two concentrations of Cd and Fe? Justify
## Which experimental design was actually adopted?
2.8. Data analysis:
Which ANOVA was applied? How many factors have you considered?
Results: English language quality can be improved.
Figures: Seems to be ok. However, a second check from Editor’s behalf will be required.
Both Cd and Fe are redox-active elements, abundant in paddy field soil. Adequate discussion from redox point of view is evident. Authors should pay due attention to this.
** Phytotoxicity of Cd and Zn on three popular Indian mustard varieties during germination and early seedling growth. Biocatalysis and Agricultural Biotechnology, 21, p.101349.
** Iron (Fe 3+)-mediated redox responses and amelioration of oxidative stress in cadmium (Cd 2+) stressed mung bean seedlings: a biochemical and computational analysis. Journal of Plant Biochemistry and Biotechnology, pp.1-12.
## Authors can improve their discussion section on ROS metabolism by reading the above mentioned articles.
Beside these, translocation behavior of both Cd and Fe need to be very concise and role of Fe-plaque formation too.
Conclusion: Ok but how the farmers will be benefited from this outcome should be included.
In my opinion, the article has some novel aspect. However, suffers from some serious issues also, need to be addressed before further consideration.
Author Response
Reviewer 2
TITLE: Influence of iron plaque on accumulation and translocation of cadmium by rice seedlings: variation of soil type and rice cultivar
Comment: ## Title change suggested. Title has to be concise yet attractive.
Response: We appreciate reviewer’s suggestion. We have shortened the title as suggested by reviewer. The title reads as below
Influence of iron plaque on accumulation and translocation of cadmium by rice seedlings
Comment: Authors sequence need to be addressed properly.
Response: Corrected.
Abstract:
Line No.20: “influence of Fe addition” amendment / supplementation is more appropriate.
Response: Replaced as per reviewer’s suggestion (please line 21, track change mode).
Comment: In general abstract is considered as a gist of the entire article in a very precise manner. A good abstract should have the experimental design and some of the important data or statistical findings. ## Rewriting suggested.
Response: We have modified the abstract with study findings as per reviewer’s suggestion.
Comment: Keywords: Ok
Response: Thank you.
Introduction:
Comment: Line No. 66: “Unlike other aquatic plants” Justify and/ change suggested.
Response: We have modified this sentence in the revised manuscript (please see lines 76-77).
Comment: While writing an article on metallic toxicant, it is expected to have permissible limits and recent evidence on similar interaction study in the introduction section.
Response: We have slightly modified the Introduction section as per reviewer’s suggestion (please lines 55-60 and 74-75).
- Materials and methods
2.1. Soil collection, processing and preparation
Comment: ## which soil sampling method did the authors actually followed?
Response: We are sorry for the confusion. Please note that each soil (about 1 ton) was collected using from a single sampling point at 4m x 4m grid area (lines 121-122, track change mode). We have included this in the revised manuscript (Section 2.1).
Comment: GCER-full name
Response: Provided (please lines 133-134, track change mode).
Comment: ## Capacity of pots?
Response: Total capacity of pot was 9 litres and 6 kg of soil was used for rice growth experiment (please see lines 139-140, track change mode).
Comment: ## Is there any particular reason behind choosing these two concentrations of Cd and Fe? Justify.
Response: Please note that in Australia, the ecological urban soil investigation level is 3 mg kg-1 although background Cd in soil is 1 mg kg-1. In Japan, the Cd in soil ranged from 0.144-3.76 mg kg-1 (Makino et al. 2016). It should be also noted that preliminary germination tests showed that rice seedling germination rates were greatly impacted at Cd levels of 5 mg kg-1. So we have chosen Cd levels up to 3 mg kg-1 in our experiment. In case Fe, as per literature most of the studies applied Fe levels of 1 and 2 g kg-1, hence we have also chosen the values as per literature (please lines 166-169).
Makino, T., Nakamura, K., Katou, H., Ishikawa, S., Ito, M., Honma, T., Miyazaki, N., Takehisa, K., Sano, S., Matsumoto, S. and Suda, A., 2016. Simultaneous decrease of arsenic and cadmium in rice (Oryza sativa L.) plants cultivated under submerged field conditions by the application of iron-bearing materials. Soil Science and Plant Nutrition, 62(4), pp.340-348.
Comment: ## Which experimental design was actually adopted?
Response: We have utilized Randomized Complete Block Design (please see line 141, track change mode).
Comment: 2.8. Data analysis:
Which ANOVA was applied? How many factors have you considered?
Response: We have employed three-way full factorial ANOVA. We have considered 3 factors for each soil type (section 2.8).
Comment: Results: English language quality can be improved.
Response: The manuscript has been checked by a native English speaker (Mr Phil Thomas, English language editor) as suggested by reviewer.
Comment: Figures: Seems to be ok. However, a second check from Editor’s behalf will be required.
Response: Thank you for the comment.
Comment: Both Cd and Fe are redox-active elements, abundant in paddy field soil. Adequate discussion from redox point of view is evident. Authors should pay due attention to this.
** Phytotoxicity of Cd and Zn on three popular Indian mustard varieties during germination and early seedling growth. Biocatalysis and Agricultural Biotechnology, 21, p.101349.
** Iron (Fe 3+)-mediated redox responses and amelioration of oxidative stress in cadmium (Cd 2+) stressed mung bean seedlings: a biochemical and computational analysis. Journal of Plant Biochemistry and Biotechnology, pp.1-12.
Response: We have discussed this in the revised manuscript (please lines 443-448, track change mode). The text reads as below
In addition to restrict Cd accumulation in plant, Fe amendment demolished of ROS and minimize oxidative stress and enhance the plant growth (Mazumder et al. 2021). Cadmium is a redox inactive heavy metal that generates ROS through indirect processes such as by triggering the Nicotinamide adenine dinucleotide phosphate (NADPH) oxidase function in the plant tissue, which eventually resulted in oxidative stress in plant tissue (Mazumder et al. 2021).
Comment: ## Authors can improve their discussion section on ROS metabolism by reading the above mentioned articles.
Response: We have discussed this in the revised manuscript (please see lines 448-456, track change mode). The text reads as below
Several studies reported that Cd has significant impact on ROS production in various plant species (Mazumder et al. 2021; Chowardhara et al. 2019). Cadmium stress increased the ROS levels including hydrogen peroxide and superoxide, which enhance the lipid peroxidation rate and ultimately destroy the antioxidant defence metabolism in plant cell in absence of Fe. Significant increase in malondialdehyde (MDA) levels were also observed in plant (Mazumder et al. 2021). So, addition of Fe supplement may reduce the MDA levels. Thus, in this present, Fe application (1 and 2 gkg-1) significantly reduced Cd accumulation in plant resulted in increased the biomass production due to improvement of antioxidant defence mechanism through scavenging ROS metabolism.
Comment: Beside these, translocation behavior of both Cd and Fe need to be very concise and role of Fe-plaque formation too.
Response: We have reduced this part briefly as per reviewer’s suggestion (please lines 390-397, 413-415).
Comment: Conclusion: Ok but how the farmers will be benefited from this outcome should be included.
Response: We have outline benefits in conclusion section (please see lines 496-499).
Reviewer 3 Report
Rice is the major cadmium (Cd) exposure pathway from agricultural soils to humans and must be reduced in an effective way. This study used a pot experiment to investigate the effectiveness of Fe application in decreasing Cd bioavailability via two types of topsoils and rice cultivars. It is a meaningful strategy to regulate grain Cd accumulation of rice plants. It is fairly good written but perhaps needs polishing.
Introduction:
Line 99-101: please concise this sentence.
Materials and methods:
Line 113-117: Did you measure any differences in levels of Fe between two soil types? And other trace metals?
Line 132-134: Has seeds germination test been conducted before the experiment?
Line 146-149: Could you please explain the reason to choose the level of Fe and Cd to add in soil.
Line 146-156: What’s the limit for Cd been regulated in agricultural soil quality in Australia?
Line 194-196: On basis of the description, this study used two-way ANOVA to identify the impacts of Fe and Cd addition. However, the experiment tested two types of soils and plants as well. Could you please clarify how to analyze the statistically significance of biomass et al. between soil types and plant species?
Results:
Line 265: Please clarify here if you check the statistical assumptions of linear regression model for your data showed in Fig 4.
Author Response
Reviewer 3
Comment: Rice is the major cadmium (Cd) exposure pathway from agricultural soils to humans and must be reduced in an effective way. This study used a pot experiment to investigate the effectiveness of Fe application in decreasing Cd bioavailability via two types of topsoils and rice cultivars. It is a meaningful strategy to regulate grain Cd accumulation of rice plants. It is fairly good written but perhaps needs polishing.
Response: We appreciate reviewer for the excellent comment. The text has been checked by a native English speaker to avoid any error.
Introduction:
Comment: Line 99-101: please concise this sentence.
Response: Done as per reviewer suggestions (please see lines 108-114)
Comment: Materials and methods:
Line 113-117: Did you measure any differences in levels of Fe between two soil types? And other trace metals?
Response: Yes, we have analyzed arsenic, Cd and Fe levels in both soils. The values were provided in the revised manuscript (please see lines 127-130).
Comment: Line 132-134: Has seeds germination test been conducted before the experiment?
Response: Yes, we have tested seed germination tests using various levels of Cd up to 5 mg kg-1, and we observed that there very little impact on rice seedlings up to 3 mg kg-1. Rice seedling growth at Cd levels of 5 mg kg-1 were adversely affected.
Comment: Line 146-149: Could you please explain the reason to choose the level of Fe and Cd to add in soil.
Response: Please note that in Australia, the ecological urban soil investigation level is 3 mg kg-1 although background Cd in soil is 1 mg kg-1. In Japan, the Cd in soil ranged from 0.144-3.76 mg kg-1 (Makino et al. 2016). It should be also noted that preliminary germination tests showed that rice seedling germination rates were greatly impacted at Cd levels of 5 mg kg-1. So we have chosen Cd levels up to 3 mg kg-1 in our experiment. In case Fe, as per literature most of the studies applied Fe levels of 1 and 2 mg kg-1, hence we have also chosen the values as per literature (please lines 166-169, track change mode).
Makino, T., Nakamura, K., Katou, H., Ishikawa, S., Ito, M., Honma, T., Miyazaki, N., Takehisa, K., Sano, S., Matsumoto, S. and Suda, A., 2016. Simultaneous decrease of arsenic and cadmium in rice (Oryza sativa L.) plants cultivated under submerged field conditions by the application of iron-bearing materials. Soil Science and Plant Nutrition, 62(4), pp.340-348.
Comment: Line 146-156: What’s the limit for Cd been regulated in agricultural soil quality in Australia?
Response: Please note that Australia has adopted strategy to maintain safe levels of cadmium in agricultural soils and crops under “The National Cadmium Minimization Strategy”. In Australia, natural levels of cadmium in the soil are low by world standards. Phosphate fertiliser has been a major source of cadmium additions to agricultural soil in Australia. The Australian fertiliser industry has made significant reductions in the cadmium contents in fertilisers over the last 20 years. It now uses rock phosphate with lower cadmium concentrations for local manufacture although the practice of adding sewage biosolids and green wastes to soils in Australia through recycling has also contributed to cadmium levels in recent years. However, cadmium levels in Australian soils are very low compared to China and other Asian countries. For example, concentrations of Cd in cultivated soils ranged from 0.11 to 6.37 mg kg-1 (median: 0.85 mg kg-1 and mean: 1.3 mg kg-1), whereas in the background (unfertilised) soil, Cd ranged from 0.02 to 1.99 mg kg-1 (median: 0.04 mg kg-1 and mean: 0.36 mg kg-1) in the Sydney region (Jinadasa et al. 1999). The ANZ environmental investigation guideline for soil Cd is 3 mg kg-1 (Jinadasa et al. 1997). Ecological urban soil investigation level is 3 mg kg-1 although background Cd in soil is 1 mg kg-1 (https://esdat.net/Environmental%20Standards/Australia/NEPM%20Tables.pdf).
We have included some of these values in the revised manuscript.
Jinadasa, N., 1999. Cadmium levels in soils and vegetables of the greater Sydney region, Australia.
Jinadasa, K.B.P.N., Milham, P.J., Hawkins, C.A., Cornish, P.S., Williams, P.A., Kaldor, C.J. and Conroy, J.P., 1997. Survey of cadmium levels in vegetables and soils of greater Sydney, Australia (Vol. 26, No. 4, pp. 924-933). American Society of Agronomy, Crop Science Society of America, and Soil Science Society of America.
Comment: Line 194-196: On basis of the description, this study used two-way ANOVA to identify the impacts of Fe and Cd addition. However, the experiment tested two types of soils and plants as well. Could you please clarify how to analyze the statistically significance of biomass et al. between soil types and plant species?
Response: We have used three-way full factorial ANOVA and significance of tests was at the 0 .05 or 0.01 level. Means and standard error (SE) were estimated from the fitted model and means were compared according to the Student’s t test (please see section 2.8).
Results:
Comment: Line 265: Please clarify here if you check the statistical assumptions of linear regression model for your data showed in Fig 4.
Response: We thank reviewer for the comment. Please note that the assumptions associated with linear regression model was checked by linearity and r2 value (correlation coefficient) in the bivariate scatter plot. We have included this in the Figure caption.
Round 2
Reviewer 2 Report
Significant improvement can be seen in R1.
Neither citing, Mazumder et al., 2021, Chowardhara et al.,
2019 is not mandatory not mandatory. It was advised to the authors to present the ROS section in better manner. So if you or authors feel, you may delete those two references.
Author Response
We are thankful to the reviewer. We have cited both references in the revised manuscript. We would like to keep both references (line 426) as reviewer has left the decision on us. So, no changes are required in the text.
Reviewer 3 Report
The authors have improved the manuscript in line with the recommendations of the reviewer.
Author Response
We highly appreciate reviewer comment.